# Change of Heart: Can Artificial Intelligence Transform Infective Endocarditis Management?

**DOI:** 10.3390/pathogens14040371

**Published:** 2025-04-09

**Authors:** Jack W. McHugh, Douglas W. Challener, Hussam Tabaja

**Affiliations:** Division of Public Health, Infectious Diseases, and Occupational Medicine, Mayo Clinic, Rochester, MN 55905, USAtabaja.hussam@mayo.edu (H.T.)

**Keywords:** infective endocarditis, artificial intelligence, machine learning, deep learning, machine vision, diagnostic imaging, echocardiography, clinical decision support, risk stratification

## Abstract

Artificial intelligence (AI) has emerged as a promising adjunct in the diagnosis and management of infective endocarditis (IE), a disease characterized by diagnostic complexity and significant morbidity. Machine learning (ML) models such as SABIER and SYSUPMIE have demonstrated strong predictive accuracy for early IE diagnosis, embolic risk stratification, and postoperative mortality, surpassing traditional clinical scoring systems. In imaging, AI-enhanced echocardiography and advanced modalities like FDG-PET/CT offer improved sensitivity, specificity, and reduced inter-observer variability, potentially transforming clinical decision making. Additionally, AI-powered microbiological techniques, including MALDI-TOF mass spectrometry combined with ML and neural network-based metagenomic classifiers, show promise in rapidly identifying pathogens and predicting antimicrobial resistance. Despite encouraging early results, widespread adoption faces barriers, including data limitations, interpretability issues, ethical concerns, and the need for robust validation. Future directions include leveraging generative AI as clinical consultative tools, provided their capabilities and limitations are carefully managed. Ultimately, collaborative efforts addressing these challenges could transform IE care, enhancing diagnostic accuracy, clinical outcomes, and patient safety.

## 1. Introduction

Infective endocarditis (IE) remains a clinically challenging infection associated with significant morbidity and an approximate mortality rate of 17% despite modern diagnostic and therapeutic advances [1]. Ongoing challenges in IE diagnosis and management persist, driven by heterogeneous clinical presentations and frequent severe complications. Artificial intelligence (AI), particularly machine learning (ML), offers new tools to improve IE care by analyzing complex clinical data, images, and genomic information.

In this review, we examine AI applications in IE across the care continuum—from early identification and imaging diagnostics to microbiologic analysis, therapy optimization, and healthcare system improvements—integrating evidence from IE-specific studies and analogous applications in related cardiovascular and infectious diseases (Figure 1). An overview and explanation of AI-related terminology used in this paper is included in Table 1. We maintain a clinical focus, highlighting how these AI tools might augment decision making for clinicians and improve patient outcomes while also discussing future directions and challenges for integrating AI into IE management. In Table 2, we highlight various AI tools that have been developed for the diagnosis and management of IE. These tools are discussed in further detail in this review.

## 2. AI for Clinical Diagnosis and Prognostication in Infective Endocarditis

### 2.1. Early Diagnosis and Risk Scores

Early identification and accurate risk stratification of IE are pivotal as delays in diagnosis or the underestimation of disease severity can lead to increased mortality and preventable complications [2]. Over the past decade, several clinical models have utilized traditional ML techniques, such as multivariable logistic regression, to better predict the risk of IE. For instance, in patients with *Staphylococcus aureus* bloodstream infection (BSI), risk-scoring systems can help clinicians determine which patients require transesophageal echocardiography (TEE). These clinical tools, such as the PREDICT and VIRSTA scores, assign point values to specific risk factors, identifying patients at low risk of IE [3,4]. In one validation cohort, a VIRSTA score of <3 demonstrated a negative predictive value (NPV) of approximately 98%, safely allowing clinicians to avoid unnecessary TEE in low-risk individuals [5]. Building upon these classical methods, Lai et al. (2024) developed the ML-based “SABIER” risk score using a random forest ML algorithm applied to an 11-year dataset encompassing over 15,000 episodes of *S. aureus* BSI [6]. The SABIER score achieved an area under the receiver operating characteristic (AUC-ROC) curve of approximately 0.74 and maintained a similarly high NPV (~98%); the AUC-ROC quantifies a model’s ability to discriminate between true positives and false positives across a range of thresholds. Scores of greater than 0.7 are considered to indicate good performance. Unlike VIRSTA, SABIER excludes subjective clinical assessments. Although external validation of the SABIER score is pending, this approach illustrates the potential role of new ML techniques in enhancing early diagnosis.

### 2.2. Predicting IE Complications

Beyond diagnosing IE itself, AI is being used to predict major complications and outcomes at an early stage. Embolic events, particularly ischemic stroke, are a dreaded complication of IE. In current practice, a vegetation size of >10 mm in echocardiography is often used as a criterion for early surgery to prevent embolism [7,8]. However, size alone may be a sub-optimal marker of embolic risk; patients with large vegetations may sometimes undergo surgery unnecessarily, while others with smaller vegetations suffer embolic strokes [9]. To improve this, researchers are applying AI to stratify embolic risk more accurately. For instance, an ongoing project in Spain is using machine vision on echocardiographic images to identify morphological features (beyond just diameter) that portend a high embolism risk [10]. Similarly, ML could combine multiple clinical variables or electrocardiogram (ECG) findings to predict other complications, such as abscess formation, heart block, or heart failure, as has been carried out for other cardiovascular diseases [11,12].

### 2.3. AI-Based Prognostication

AI-based risk stratification has also shown promise in predicting mortality from the point of diagnosis and following cardiac valve surgery. One study applied ML to a panel of 27 inflammatory cytokines in IE patients. It identified IL-15 and CCL4 levels as key predictors of mortality, achieving ~91% accuracy for death prediction when combined with C-reactive protein [13]. Such biomarker-based ML models could flag patients for more aggressive therapy or closer monitoring early in their course. Another study developed and validated a risk model (“SYSUPMIE”) using an XGBoost ML algorithm to predict early mortality following surgery for IE [14]. The model incorporated eight clinical variables, achieving good predictive accuracy (AUC: ~0.81) in both internal and external validation cohorts, outperforming traditional models like EuroSCORE II. An open-access online calculator has been made available to facilitate practical clinical use and decision making.

In summary, AI-driven risk stratification can assist clinicians in the early recognition of IE and in prognostication. Early studies are encouraging, but larger prospective validation is needed before such models can be reliably incorporated into routine practice.

## 3. AI in Diagnostic Imaging

### 3.1. Echocardiography

Echocardiography remains the primary imaging modality for diagnosing IE; however, its diagnostic accuracy may vary with operator expertise and can be limited by patient anatomy. False negatives occur, especially in patients with prosthetic valves or poor acoustic windows, and artifacts can be mistaken for vegetations [15]. AI can assist by improving image quality, analysis consistency, and highlighting subtle features. A 2024 study by Sineglazov et al. developed a DL machine vision system to automatically segment and detect valvular vegetations in echocardiographic images [16]. The system achieved a high accuracy in identifying and calculating the volume of IE vegetations. By analyzing pixel-by-pixel information across the cardiac cycle, such models can potentially distinguish true infective lesions from motion artifacts or normal structures. In practice, this means an AI-enhanced echocardiogram could, for example, alert a physician to a small vegetation that human readers might miss or confirm that an apparent mass is likely an artifact. Preliminary reports indicate that incorporating AI leads to more standardized and sensitive echo interpretation across various cardiac diseases, which is especially valuable when TEE is equivocal [17,18]. Future ML models may also take advantage of large-scale data extraction from echocardiography reports using NLP to generate insight into the risk of IE and prognostication. This approach has been shown to be feasible in one large study [19].

### 3.2. Advanced Imaging

AI is also being applied to advanced imaging modalities for IE. Cardiac computed tomography (CT) is often used as a complementary tool when echocardiographic findings are inconclusive or when detailed anatomic delineation is needed (e.g., assessing abscesses or aneurysms) [20]. While CT provides high-resolution anatomical detail, interpreting subtle peri-valvular changes in CT can be challenging. AI algorithms can mine CT data for patterns imperceptible to radiologists, for instance, differentiating sterile postoperative changes from true abscess cavities around a prosthetic valve [21]. AI could also facilitate multimodal integration, aligning echo and CT images and jointly analyzing them to spot concordant abnormalities (e.g., a valve defect in CT corresponding to a vegetation in an echo). Early work in this area suggests improved sensitivity for structural complications of IE by leveraging the strengths of each modality [22].

For nuclear imaging studies, like positron emission tomography with CT (FDG-PET/CT), AI has demonstrated enhancements in diagnostic performance. These scans are especially useful in prosthetic valve endocarditis (PVE) and device-related IE, where metabolically active infection foci can be visualized. Traditionally, the interpretation of PET relies on the subjective analysis of uptake patterns, but AI can add quantitative rigor. In one proof-of-concept study, ML algorithms combined the results of PET/CT with other diagnostic components of the modified Duke/ESC 2015 criteria [23], achieving higher diagnostic specificity without compromising sensitivity. Of four ML model approaches, an ensemble model had the highest AUC of 0.41 versus 0.917 for the conventional criteria [24]. In another 2023 study, a radiomics-based ML approach to FDG-PET/CT in suspected PVE was able to increase the sensitivity of diagnosis from 59% to 72% [25]. In other words, the AI analysis of PET images identified infectious signatures that human readers might overlook, leading to more cases being correctly recognized as IE. Another nuclear technique, leukocyte single-photon emission computed tomography/computed tomograph (SPECT/CT), had higher specificity than PET for detecting IE in some series [26]. Embedding these imaging techniques into AI-driven pipelines could further improve the detection of IE in challenging cases.

In summary, AI in imaging could improve the detection of vegetations and complications (abscesses, fistulae, or device infections) by reducing operator variability and enhancing sensitivity. It can differentiate pathology from artifacts in echocardiograms, integrate multimodal imaging data for a more comprehensive assessment, and analyze functional imaging (PET/SPECT) with greater objectivity. As these techniques mature, we anticipate an AI “co-pilot” alongside cardiologists and radiologists, for example, automatically flagging an enlarging abscess in serial CTs or calculating vegetation growth over time in serial echoes.

## 4. Microbiological and Genomic Applications

### 4.1. Pathogen Identification

Microbiological diagnosis is another arena where AI is making inroads, from pathogen identification to prediction of antimicrobial resistance. For instance, advanced ML algorithms have been integrated with matrix-assisted laser desorption ionization–time-of-flight (MALDI-TOF) mass spectrometry and associated matching software to enhance identification precision. Although traditional MALDI-TOF approaches, which inherently depend on reference libraries and software matching, reliably identify organisms at the genus level, accuracy can diminish for closely related species. ML can analyze subtle differences in mass spectral patterns. In one study on *Brucella* species, an ensemble of ML classifiers achieved 100% accuracy in differentiating three *Brucella* species based on their MALDI spectra, whereas standard methods struggled to go beyond genus-level identification [27]. This kind of approach could be applied to other IE pathogens, for example, distinguishing *Streptococcus gallolyticus* from other *streptococci*, or identifying unusual causes of endocarditis that MALDI databases might misclassify. While the clinical benefit of these types of applications remains uncertain, leveraging ML-enhanced pathogen identification could yield important epidemiological insights, streamline outbreak detection, and ultimately improve clinical outcomes through more precise microbial diagnostics.

### 4.2. Predicting Antimicrobial Resistance

Beyond identification, AI is being used for predictive analytics in antimicrobial resistance (AMR). ML models can learn from genotype–phenotype datasets to predict an organism’s antibiotic resistance profile without waiting for traditional susceptibility tests. For instance, AI can interpret whole-genome sequencing data or mass spectrometry data of a bacterium to infer if it carries resistance genes or mutations, thereby guiding therapy earlier. In one demonstration, an AI model using MALDI-TOF outputs successfully predicted drug resistance in *Pseudomonas aeruginosa*, a pathogen relevant to healthcare-associated IE [28].

AI is also poised to assist in the microbiological diagnosis of blood culture-negative endocarditis (BCNE) and rare causes of IE. When cultures are negative, clinicians must rely on serologic tests (for organisms like *Coxiella burnetii* or *Bartonella*) or newer methods like polymerase chain reaction (PCR) and sequencing of valve tissue [29]. These approaches generate complex data; for example, metagenomic sequencing of an excised heart valve can yield millions of reads of DNA. Novel neural network-based tools, such as transformer-based classifiers, can help analyze this large volume of data, identifying pathogen-derived sequences even when they are absent from reference databases, thus improving diagnostic accuracy and expediting the detection of rare or previously unrecognized pathogens causing BCNE [30].

## 5. Treatment and Care Coordination

Managing IE often requires complex decisions, such as choosing the appropriate antibiotic regimen, optimizing dosing, and ensuring adherence to evidence-based guidelines over a prolonged course of treatment. AI can support clinicians in making these therapeutic decisions more personalized and data-driven. By learning from large datasets of IE cases, AI systems could recommend interventions optimized for each patient, potentially improving outcomes across the board.

### 5.1. Appropriate Antibiotic Selection and Dosing

One area of interest is AI-driven antibiotic selection and dosing optimization. Treatment of IE typically entails high-dose, prolonged antibiotic therapy, and getting it right is critical to eradicate infection while minimizing toxicity. Decision-support algorithms are being developed to integrate patient-specific data, e.g., the organism identity and minimum inhibitory concentration (MIC), local resistance patterns, kidney function, allergies, and the presence of prosthetic material, to suggest the best antibiotic regimen and dosage [31]. Precision Rx^®^, DoseMeRx^®^, InsightRX^®^, and PrecisePK^®^ are case examples of tools that integrate Bayesian statistics with ML models to optimize precision dosing. Extending this, ML models could continuously adjust dosing in response to patient drug levels and renal function trends or even predict the ideal duration of therapy needed. While formal studies in IE are limited, analogous work in sepsis management shows promise. For instance, researchers used AI to analyze ICU patients’ vitals and labs in real time and provide recommendations on the timing of antibiotics. When clinicians followed the AI-suggested timing, the 30- and 60-day sepsis mortality was lower [32]. In IE, where the treatment duration is typically a fixed 4–6 weeks, AI might, one day, help identify patients who could safely receive shorter courses by recognizing patterns in their response to therapy. It could also prevent errors like premature discontinuation by ensuring that guideline-recommended durations are met through an EHR alert system.

### 5.2. Decision Support and Care Coordination

AI can further enhance clinical management through decision support and care coordination. IE management spans multiple specialties (cardiology, cardiac surgery, infectious diseases, neurology, etc.), and adherence to guidelines (such as obtaining prompt ID consult, follow-up blood cultures, or early surgical evaluation when criteria are met) is crucial. However, real-world practice varies. Surveys show that in many hospitals, not every IE patient obtains an ID specialist consult or even an echocardiogram [33]. An AI-powered clinical decision support system integrated into the EHR could improve adherence by triggering reminders or checklists. Similarly, it could alert providers if an IE patient with indications for surgery has not been reviewed by a surgeon, or if recommended ancillary tests (e.g., dental evaluation) were overlooked. By parsing clinical notes with NLP, such a system might also catch undocumented findings. Indeed, NLP algorithms have outperformed ICD code searches in identifying specific patient cohorts; for instance, an NLP+ML approach was better than billing codes at finding patients who inject drugs (PWIDs) by analyzing clinical notes [34]. This suggests AI could identify at-risk IE populations and ensure they receive early intervention.

Patients with IE typically receive a period of inpatient care and are then dismissed to the outpatient setting with intravenous or oral antibiotics for a 4–6-week course. Incorporating ML tools into outpatient parenteral antimicrobial therapy (OPAT) could enable risk stratification for adverse events, facilitate early identification of clinical deterioration, and optimize antimicrobial management by predicting medication intolerance or toxicity. Moreover, ML algorithms could assist in resource allocation, enhancing patient monitoring efficiency and potentially reducing unplanned hospital readmissions [35]. Overall, AI has the potential to function as a safety net and a consultant in the background, enhancing guideline adherence and providing point-of-care recommendations tailored to each complex IE case.

### 5.3. Multi-Disciplinary Teams

A key aspect of IE management is the expert input from a multi-disciplinary team (MDT), composed of radiologists, cardiovascular surgeons, infectious disease physicians, pharmacists, and nursing staff. IE MDTs, recommended by both the European Society of Cardiology and the American Heart Association, have been shown to improve patient outcomes through coordinated expertise [36]. However, MDTs inherently grapple with limitations, such as variability in clinical judgment and potential biases of team members. AI could enhance MDT’s effectiveness by synthesizing complex patient data (clinical histories, imaging, and microbiologic profiles) into streamlined, actionable summaries through generative AI models, thus facilitating more informed consensus-based decisions. RL algorithms, leveraging historical treatment outcomes, might also dynamically optimize critical MDT decisions, such as timing for surgical interventions or strategies to mitigate embolic risk. Such integration of AI could standardize and elevate MDT recommendations, ensuring more consistent evidence-based care for complex IE management scenarios.

## 6. Challenges and Limitations

While AI holds substantial promise for IE, several barriers must be addressed before integration into clinical practice is feasible.

### 6.1. Data Quality and Quantity

Most of the IE datasets used to generate ML models discussed in this paper were single-center, increasing the risks of overfitting and poor generalizability. IE is a rare diagnosis; hence, large, standardized multicenter datasets from collaborative registries will be needed to optimize models, as has been demonstrated elsewhere [37]. International cooperation in data sharing could accelerate the development and validation of robust models.

### 6.2. Scalability

Beyond issues of dataset quality and size impacting generalizability, a significant challenge remains in the scalable implementation of AI-driven management strategies across diverse healthcare settings. Variability in infrastructure, resources, and technical expertise among hospitals can hinder the broader distribution and adoption of sophisticated AI tools initially developed in specialized centers. To mitigate this, strategies such as standardized integration into widely used electronic health records or leveraging cloud-based AI solutions should be considered. These approaches could enhance accessibility, ensure model consistency across settings, and facilitate smoother workflow integration, ultimately promoting the wider generalizability and clinical utility of AI-driven management for IE.

### 6.3. Model Interpretability and Trust

Many ML models function as “black boxes”, limiting clinical trust due to their opaque decision-making processes. This opacity may lead to clinician disengagement or "alert fatigue", reducing the effective use of potentially valuable predictive insights. Additionally, prognostication models must consider clinical utility, particularly the modifiability of identified risk factors [38]. Prognostic tools identifying predominantly non-modifiable factors offer limited practical value, as clinicians cannot alter patient trajectories based on these insights. Ensuring interpretability and emphasizing actionable, modifiable risks could foster clinician acceptance, mitigate risk score fatigue, and, ultimately, enhance patient outcomes. Continuous collaboration with clinicians during model development is essential to ensure meaningful integration into clinical workflows and sustained utility in practice [39].

### 6.4. Integration into Workflow

Seamless integration into existing electronic health records and imaging systems is crucial. User-centered design, careful timing of AI prompts, and clinician training are required to avoid workflow disruptions. Pilot testing in real-world clinical environments will be key to successful adoption. Indeed, crossing the “research-to-practice” chasm has been challenging across many medical fields. Numerous predictive models have been rigorously studied, yet very few have achieved meaningful clinical implementation; this challenge is not unique to IE.

### 6.5. Validation and Regulatory Hurdles

Rigorous prospective validation is necessary to meet regulatory requirements and demonstrate improved clinical outcomes. Evolving oversight frameworks must adapt to the dynamic nature of AI, highlighted by previous controversies like the UK’s DeepMind Health project, which faced criticism over patient data handling [40]. Clear guidance from regulatory bodies will help institutions confidently deploy AI tools.

### 6.6. Ethical and Bias Concerns

AI models may inadvertently reinforce biases present in training data, affecting certain patient groups disproportionately [41,42]. Defined accountability, robust ethical guidelines, and maintaining human oversight are essential to prevent automation bias and ensure equitable care. Regular auditing of AI models for bias and fairness is necessary to maintain clinical trust [43].

## 7. Future Directions

### 7.1. Novel Therapeutics

AI is expediting the discovery of new antimicrobial agents. ML models can screen vast chemical libraries to identify novel compounds with activity against a variety of pathogens. In a landmark example, a DL approach identified a completely new broad-spectrum antibiotic (“halicin”), which is structurally distinct from existing antibiotics and capable of killing a wide range of drug-resistant bacteria (including *Mycobacterium tuberculosis* and carbapenem-resistant Enterobacteriaceae) [44]. Indeed, the integration of generative AI in drug design has progressed to the point that the first fully AI-designed drug has entered human clinical trials as of 2023 [45], heralding a new era where algorithms generate candidate molecules for testing. While that drug is not for endocarditis, similar AI-driven pipelines are being applied to discover novel antibiotics, antifungals, and antivirals targeted at urgent infectious threats.

### 7.2. Generative AI as Consultative Co-Pilot

This review has primarily focused on the application of ML and DL models for optimizing diagnostic accuracy and prognostication. However, the past two years have seen rapid advancements in generative AI, particularly regarding large language models (LLMs), which are often refined using RL to improve their functionality. The utility of generative AI is likely to be manifold. Recent studies have demonstrated that these models can already surpass medical students on licensing examinations [46] and outperform ID fellows and attendings in solving ID-specific clinical problems [47]. Additionally, LLMs have achieved near-specialist-level performance when providing consultative guidance for managing BSIs [48]. Given ongoing enhancements in computational power and efficiency, it is conceivable that LLMs may eventually surpass specialists in delivering comprehensive consultative recommendations.

Despite these promising developments, integrating generative AI into clinical care for IE carries significant risks. Current LLMs exhibit limitations, including a lack of genuine contextual awareness, tendencies to generate inaccurate information (i.e., confabulation), and the amplification of biases inherent in their training data. These factors pose considerable threats to patient safety if such technologies are prematurely implemented or inadequately validated. Thus, careful and deliberate steps, along with a thorough understanding of both the strengths and limitations of generative AI, are essential for its responsible integration into IE management [49]. Given these concerns, AI currently serves as a tool to augment clinical decision making rather than replace specialist consultation. Whether advancements in AI will ultimately alter this role remains uncertain, underscoring the need for continued scrutiny and responsible integration as the technology evolves.

## 8. Conclusions

AI shows considerable promise for improving the diagnosis, risk stratification, and management of IE through enhanced clinical prediction models, imaging diagnostics, and microbiological techniques. However, broad implementation requires addressing challenges related to data availability, model transparency, clinical workflow integration, validation, and ethical concerns. With collaborative effort, AI can transform IE care, leading to faster diagnosis, optimized therapies, and improved patient outcomes.

## Figures and Tables

**Figure 1 pathogens-14-00371-f001:**
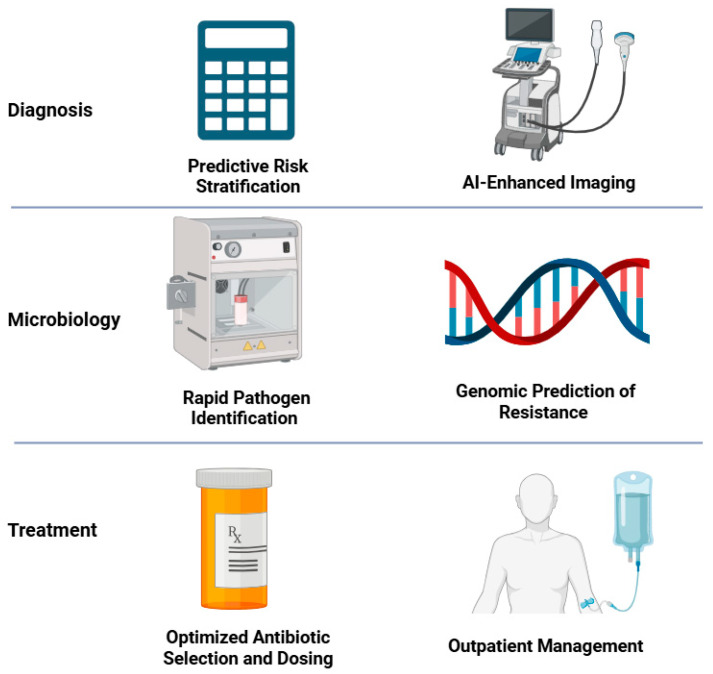
Use cases for artificial intelligence in the diagnosis and management of infective endocarditis (created with BioRender.com).

**Table 1 pathogens-14-00371-t001:** Overview of artificial intelligence tools for infective endocarditis diagnosis and management.

Type of Application	Explanation
Machine learning (ML)	ML refers to algorithms that learn from data patterns to predict outcomes or classify information.
Deep learning (DL)	DL is a specialized ML technique using neural networks, excellent at analyzing complex data such as medical images for subtle features.
Natural language processing (NLP)	NLP enables computers to understand and interpret human language, useful for analyzing medical notes or patient records.
Predictive analytics	Techniques that use historical and real-time data to forecast patient outcomes, such as risk of complications or mortality.
Reinforcement learning (RL)	RL trains AI through trial and error, guiding clinical decision making by evaluating outcomes from previous cases.
Generative AI	AI methods that can generate synthetic data or scenarios to augment datasets, helping models learn better, especially for rare diseases like IE.

**Table 2 pathogens-14-00371-t002:** Comparative overview of AI models in infective endocarditis: performance, advantages, and limitations.

AI Model/Application	Category	Accuracy/Performance	Advantages	Limitations
SABIER (ML)	Predictive	AUC: ~0.74; NPV: ~98%	Objective and eliminates subjective criteria; strong predictive capability for IE in *S. aureus* BSI	Awaiting external validation; reliance on retrospective data
SYSUPMIE (ML—XGBoost [version 1.2.1])	Predictive	AUC: ~0.81	Outperforms traditional risk models in predicting postoperative mortality; online calculator available	Limited external validation; dataset-dependent
Echocardiography image analysis (DL)	Imaging	High accuracy in vegetation detection and segmentation	Enhances diagnostic precision; reduces inter-observer variability; identifies features human observers might miss	Dependent on image quality; potential interpretability issues
FDG-PET/CT radiomics (ML)	Imaging	Improved sensitivity (59% → 72%) in PVE diagnosis	Quantitative and reduces subjective interpretation; increases diagnostic sensitivity	Requires high-quality imaging data; complexity in clinical workflow integration
ML-based MALDI-TOF identification	Microbiology	Up to 100% accuracy in differentiating closely related bacterial species	Enhances rapid pathogen identification precision; improves epidemiological tracking	Currently validated for limited pathogens; broad implementation requires extensive spectral libraries
Genomic prediction of antibiotic resistance (ML)	Microbiology	High accuracy in phenotype prediction from genomic data	Facilitates rapid therapeutic decisions by predicting antibiotic resistance profiles without culture delays	Dependent on quality and comprehensiveness of genomic databases
Generative AI (LLMs)	Clinical decision support	Comparable performance to students, ID fellows, and attendings	Rapid synthesis of extensive patient data; potential to significantly augment clinical decision making	Confabulation risk and bias amplification; lacks contextual awareness; requires careful validation
Reinforcement learning for clinical decision support (RL)	Clinical decision support	Promising, though accuracy metrics not detailed	Dynamically optimizes timing and strategy for surgical intervention and clinical management	Requires extensive training datasets; interpretability challenges

## Data Availability

No new data were created or analyzed in this study. Data sharing is not applicable to this article.

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
