# Peer review of "Change of Heart: Can Artificial Intelligence Transform Infective Endocarditis Management?"

_pathogens, 2025, doi:10.3390/pathogens14040371_

Round 1
Reviewer 1 Report
Comments and Suggestions for Authors
The manuscript is a review. It provides a broad look at the emerging interfaces where artificial intelligence (AI) ranging from machine learning (ML) and deep learning (DL) to generative AI and reinforcement learning (RL) can impact multiple facets of infective endocarditis (IE) including microbiologic and imaging diagnosis, prediction of complications, prognostication of outcome, and aspects of day- to -day patient management in the hospital and out-patient setting. Both the potential benefits of “AI”, but also its limitations including that of databases for analysis, and even the regulatory and continued need for clinician in-put requirements are noted. From the studies cited it seems clear that the areas of greatest progress to date are those where in highly focused investigators can direct their attention and seek advances through technology enhanced by ML and DL - such as improved interpretation of FDG-PET/CT or where clinical data can be more intensively analyzed for predicting outcome. The manuscript is thoughtful and clearly written.
With the increasing dispersion of electronic medical records and data recording and the digitization of imaging it seems that medicine in the developed world is at the point where AI could benefit care of patients with endocarditis, as well as many other diseases more prevalent than IE. The relative rarity of IE in terms of sequential targeting AI resources in health care to address areas based on magnitude of need may be a challenge for IE and lead to piecemeal application on focused specific bits of progress by investigators and industry. Nevertheless, two things strike me as unaddressed but meriting consideration because of the potential to effect patient care in places where it currently is in great need. One is how do we distribute the broad management potential (the AI co-pilot aspect) to a wide array of hospital to impact care.
A second comes with recognition of the complexity of endocarditis management and the resulting guidance from the European Society of Cardiology and the American Heart Association to, where possible, use multispecialty endocarditis teams to manage patients with IE. Data have begun to show the benefits of this approach. This seems to be a second area where AI could truly effect IE outcome. These teams are in some respects nascent AI, attempting to sort out the most beneficial care in spite of the limitations and biases of the team members. It would be of interest, and hopefully motivation, to hear the authors thoughts on how DL, RL and generative AI can be brought together to help teams answer complex management questions such as cardiac surgery to prevent embolic complications, the role and risks of surgery to treat cardiac dysfunction or eradicate infection, etc - the tough decisions where teams need evidenced based guidance (? from AI), Some suggested approaches to this challenge might stimulate needed work.
Author Response
Reviewer 1 asked for additional comment on how the broad management potential to a wide array of hospital care. We have added a sub-section “6.2 Scalability” that discusses the challenges and opportunities associated with scaling AI solutions across hospital systems. Reviewer 1 also asks for additional details on how AI can be used to improve the care delivered by multi-disciplinary teams. In response, we have written a specific sub-section “5.3 Multi-Disciplinary Teams” addressing this.
We thank the reviewer for these thoughtful comments and are confident that the revised manuscript is improved as a result of reviewer feedback.
Reviewer 2 Report
Comments and Suggestions for Authors
The paper is a good review article where the possible role of artificial intelligence and its different tools such as Machine, Learning, Deep Learning, Natural Language Processing, Predictive Analytics, and others in the various diagnostic and therapeutic aspects of this disease is discussed. A useful overview of AI-related terminology is provided in Table 1.
The content is well organized, with logical sections covering diagnosis, prognosis, imaging, microbiology, and future challenges. Recent studies are cited and innovative tools are mentioned, which is an important effort.
Implications for early diagnosis, disease and its complications, possible improvement in detecting images compatible with Endocarditis on echocardiogram, cardiac CT, and nuclear medicine tests are discussed in detail. Likewise, the possibilities of a better microbiological diagnosis and a better selection of antimicrobial treatment are highlighted. Challenges such as data quality, biases, and clinical integration are also addressed, which brings balance to the article.
For all these reasons, I think it is a very useful article to visualize the windows of opportunity in the management of these patients with artificial intelligence.
Specific comments
Consider specifying the meaning of “AUC”, although it is a very well known acronym.
As additional elements value to add a comparative table of AI models (accuracy, advantages, limitations) for a better understanding of these tools and a flowchart on how to integrate AI in the management of IE. In particular, which IA tools are most useful at each stage of the clinical course of endocarditis.
Author Response
Reviewer 2 asks for a definition of AUC, which we have provided. Reviewer 2 also asks for a flowchart on how to integrate AI into the management of IE. In response we have updated Figure 1 to show the use cases for AI at the various stages of IE diagnosis and management. Reviewer 2 also asks for a comparative table of AI models (accuracy, advantages, limitations). We have create a table as requested and included it in the updated manuscript.
We thank reviewer 2 for the thoughtful comments and are confident that the revised manuscript is improved as a result of reviewer feedback.